# Discovery of Bis-Imidazoline Derivatives as New CXCR4 Ligands

**DOI:** 10.3390/molecules28031156

**Published:** 2023-01-24

**Authors:** Zhicheng Zhou, Isabelle Staropoli, Anne Brelot, Peggy Suzanne, Aurélien Lesnard, Fanny Fontaine, Serge Perato, Sylvain Rault, Olivier Helynck, Fernando Arenzana-Seisdedos, Jana Sopkova-de Oliveira Santos, Bernard Lagane, Hélène Munier-Lehmann, Philippe Colin

**Affiliations:** 1Viral Pathogenesis Unit, Department of Virology, Institut Pasteur, 75015 Paris, France; 2Centre d’Etudes et Recherche sur le Médicament de Normandie (CERMN), Université Normandie, UNICAEN, 14032 Caen, France; 3Unité de Chimie et Biocatalyse, Institut Pasteur, Université Paris Cité, CNRS UMR3523, F-75015 Paris, France; 4Infinity, Université de Toulouse, CNRS, INSERM, UPS, 31024 Toulouse, France

**Keywords:** CXCR4, CXCL12, bis-imidazoline, antagonist, therapeutic target, HIV-1 infection

## Abstract

The chemokine receptor CXCR4 and its ligand CXCL12 regulate leukocyte trafficking, homeostasis and functions and are potential therapeutic targets in many diseases such as HIV-1 infection and cancers. Here, we identified new CXCR4 ligands in the CERMN chemical library using a FRET-based high-throughput screening assay. These are bis-imidazoline compounds comprising two imidazole rings linked by an alkyl chain. The molecules displace CXCL12 binding with submicromolar potencies, similarly to AMD3100, the only marketed CXCR4 ligand. They also inhibit anti-CXCR4 mAb 12G5 binding, CXCL12-mediated chemotaxis and HIV-1 infection. Further studies with newly synthesized derivatives pointed out to a role of alkyl chain length on the bis-imidazoline properties, with molecules with an even number of carbons equal to 8, 10 or 12 being the most potent. Interestingly, these differ in the functions of CXCR4 that they influence. Site-directed mutagenesis and molecular docking predict that the alkyl chain folds in such a way that the two imidazole groups become lodged in the transmembrane binding cavity of CXCR4. Results also suggest that the alkyl chain length influences how the imidazole rings positions in the cavity. These results may provide a basis for the design of new CXCR4 antagonists targeting specific functions of the receptor.

## 1. Introduction

CXCR4 is a chemokine receptor belonging to the family of 7-transmembrane, G Protein Coupled Receptors (GPCRs). It is expressed in a broad range of cells, including leukocytes, hematopoietic stem cells, epithelial, endothelial and neuronal cells [1]. It is present in hematopoietic and lymphoid tissues, as well as in a variety of non-hematopoietic tissues (intestine, brain, kidney, liver, lung). Its natural ligand is the chemokine CXCL12/Stromal Cell-derived Factor-1 (SDF-1) [2,3], which also binds atypical chemokine receptor 3 (ACKR3)/CXCR7 [4,5]. Other endogenous peptide ligands of CXCR4 have been described, such as macrophage migration inhibitory factor (MIF) [6], extracellular ubiquitin [7] and human β-defensins 2 and 3 [8]. Recent data have revealed that CXCR4 also binds natural amines (histamine, dopamine, serotonin) on the surface of plasmacytoid dendritic cells, regulating type-I interferon production in these cells [9]. CXCR4 is also known to be the receptor for viral ligands, such as viral Macrophage Inflammatory Protein-II (vMIP-II) encoded by Kaposi’s sarcoma-associated herpes virus [10] and the surface subunit (gp120) of the envelope glycoprotein of X4 strains of HIV-1, the etiologic agent of AIDS [11]. CXCR4 and the related chemokine receptor CCR5 are the two main coreceptors for HIV-1 entry into CD4+ immune cells. HIV-1 gp120 binds to CD4 and then to the coreceptor, triggering fusion between the viral and cell membranes and release of the viral nucleocapsid into the cytosol [12]. CCR5-using viral strains predominate in the early phase of infection, while CXCR4-using viruses may emerge later in patients and are associated with increased pathogenicity of infection.

The CXCR4/CXCL12 couple plays critical functions during embryonic life and in adults. Mice deficient for CXCL12 or CXCR4 die perinatally, showing defects in B-cell lymphopoiesis; absence and reduced myelopoiesis in the bone marrow and liver, respectively; and abnormal development of the cerebellum, the cardiac septum and the vasculature in the gastrointestinal tract and the kidney [13,14,15,16,17]. In adulthood, CXCR4/CXCL12 regulates the migration of leukocytes, their homing to and retention in primary lymphoid organs, and their homeostasis in the periphery [18]. At the cellular level, the binding of CXCL12 to CXCR4 induces a plethora of effects, including chemotaxis (i.e., directed cell migration), proliferation, survival, adhesion and gene expression. These cellular responses result from upstream intracellular signaling pathways initiated by heterotrimeric G-proteins and β-arrestins [19,20].

From a pathophysiological standpoint, since the discovery of its role in HIV-1 infection [11], CXCR4 has been involved in a variety of diseases, including inflammatory diseases [21,22], autoimmune diseases [23], primary immunodeficiencies (e.g., the WHIM syndrome [24,25]) and several types of hematological malignancies and solid tumors [18,26], highlighting its role as a promising therapeutic target. During the last two decades, various types of molecules designed to block CXCR4/CXCL12 interactions have been developed, including small molecule compounds [27,28,29,30,31], peptides [32,33,34], monoclonal antibodies [35,36] and nanobodies [37]. Many of these molecules have been investigated in preclinical studies or in clinical trials, in particular as antitumor compounds, and showed promising effects either in monotherapy or in combination with conventional therapies (for recent reviews, see [26,38]). To date, however, the bicyclam derivative AMD3100 (Plerixafor) is the only CXCR4 inhibitor approved for clinical use. This molecule is used for mobilization of hematopoietic stem cells from the bone marrow and subsequent autologous transplantation in patients with Non-Hodgkin’s lymphoma or multiple myeloma [39,40]. Many molecules that have entered clinical trials, however, have not been continued due to poor pharmacokinetic profiles, low efficacy or too high toxicity.

Because of its wide distribution and numerous physiological functions, the question of targeting CXCR4 in long-term treatments remains a challenge and is still the subject of investigations. An ideal compound would be able to interfere with CXCR4-dependent diseases while preserving physiological functions of the receptor. In the field of HIV/AIDS, compounds have been described to exert antiviral activity at concentrations that preserve CXCR4 signaling [41,42]. In the same vein, we have recently shown that CXCR4 can exist in different conformations at the cell surface, some of which are used by particular HIV-1 strains but not by CXCL12 [43]. Similarly, some CXCR4 ligands induce different signaling responses compared to CXCL12, presumably owing to different interaction sites in the receptor binding pocket [44]. These data therefore suggest that it is conceivable to inhibit some functions of CXCR4 while preserving others.

With these considerations in mind, we have developed a high-throughput screening protocol to identify novel CXCR4 ligands. We have characterized new molecules belonging to the same chemical family from a chemical library of 6851 compounds. These molecules are bis-imidazoline compounds, having in common two hydrogenated imidazole rings linked by an alkyl chain of variable length. Imidazoline groups have been described in several drugs [45], some with antiviral activities against different virus families [46,47] including HIV [48,49]. Here, we show that some of the molecules we identified antagonize CXCL12 and anti-CXCR4 mAb 12G5 binding with submicromolar potency, inhibit CXCL12-induced chemotaxis and show anti-HIV-1 activity. Interestingly, the inhibitory potencies of molecules varied with the alkyl chain length. Docking studies of molecules to different models of the CXCR4 binding site combined with site-directed mutagenesis indicate that the alkyl chain contributes to positioning both imidazole rings within the receptor binding cavity. Changing the length of the alkyl chain alters the mode of binding of imidazole rings to the receptor, which could explain why the activity of the molecules is then modified. These results therefore open new possibilities to optimize the binding of functional groups to CXCR4 and modulate its functional properties.

## 2. Results and Discussion

### 2.1. Identification of New CXCR4 Ligands through High-Throughput Screening

As a starting point of this study, we set up a high-throughput screening of 6851 compounds available at the time from the CERMN (Centre d’Etude et de Recherche sur le Médicament de Normandie) chemical library for their ability to interact with CXCR4 (see Section 3.3 for details). We applied a homogeneous cell-based binding assay using Homogeneous Time-Resolved Fluorescence resonance energy transfer (HTRF) Tag-lite technology [50]. In brief, adherent HEK293 cells stably expressing CXCR4 fused to a Lumi4-Tb-labeled SNAP-tag (ST-CXCR4) were incubated with a red-emitting CXCL12 derivative (red-CXCL12) in the presence or absence of compounds. Red-CXCL12 was used at 20 nM, a concentration close to its equilibrium dissociation constant (K_D_) value for CXCR4 [51]. We investigated the capacity of the compounds to inhibit energy transfer between the Lumi4-Tb donor and acceptor red-CXCL12, which results in a decrease in the HTRF signal (i.e., the ratio between the acceptor and donor fluorescence signals at 665 and 620 nm, respectively). As a control for inhibition of the binding of red-CXCL12, the HTRF signal in the presence of an excess concentration (15 μM) of the CXCR4 antagonist AMD3100 was determined in parallel (and used as a negative control in each plate). In the primary screening, compounds were tested at a single dose (6.6 µg/mL corresponding to an average concentration of approximately 20 µM). Each 384-well plate was analyzed independently. First, using negative and positive controls, the value of the Z′ factor [52] was determined. All values were greater than 0.5 (mean of 0.551 ± 0.036), indicating a robust and reliable assay. The negative controls were then used to calculate the relative HTRF ratio (corresponding to the HTRF ratio of the compound divided by the HTRF ratio of the negative controls). A hit was defined as a compound exhibiting a relative HTRF ratio less than 1.3. Data analysis identified eight hits. Among them, three compounds belong to the same chemical series, namely, bis-imidazoline compounds formed by two hydrogenated imidazole rings linked by an alkyl chain of variable length (see Table 1; HTRF ratio values of 1.26, 1.17 and 1.19 for compounds mr20347, mr20349 and mr20350, respectively, containing an alkyl chain of 8, 10 or 12 carbons, respectively). These three compounds were selected for further affinity studies and biological validation. To gain insight into the structural basis for their CXCR4 binding property, different analogs with different alkyl chain length were also synthesized and included in these subsequent studies (Table 1).

### 2.2. Bis-Imidazoline Compounds Have Submicromolar Affinities for CXCR4, Which Depend on the Alkyl Linker Length

To obtain information on the structural requirements necessary for the interaction with CXCR4, we performed dilution series experiments (see Section 3.4.) by varying the concentrations of bis-imidazoline compounds in the HTRF assay used for the screening campaign (Table 1 and Figure 1A). In agreement with the primary screening, mr20347, mr20349 and mr20350 when used at high concentrations fully inhibited HTRF, similarly to AMD3100 (red symbols in Figure 1). The half-maximal inhibitory concentration (IC_50_) values for these three compounds tended to increase with increasing chain length (IC_50_ = 350, 470 and 760 nM for mr20347, mr20349 and mr20350, respectively, Table 1). These values were in the same range as the value found for AMD3100 (130 nM). The result for mr20349 was further confirmed in displacement experiments of ^125^I-labeled CXCL12 binding to endogenous CXCR4 expressed in blood mononuclear cells from a healthy donor (data not shown). Compared to these three molecules, those with an odd chain length (mr20346, mr20348, mr34200) displayed weaker inhibitory potency and efficacy, suggesting that symmetry in the molecule is required for optimal binding to CXCR4. Reduction of the alkyl linker length to n = 6 further reduced the capacity to compete with red-CXCL12 (mr20345). Finally, replacing the imidazole rings by pyrimidines (heterocycle that also contains nitrogens in positions 1 and 3 but has 4 carbons instead of 3) in some of these compounds fully abolished the inhibitory activity (data not shown), even at the highest tested concentration (25 μM), suggesting that the five-membered structure of the imidazole cycle is crucial in CXCR4 binding. Of note, most of the tested compounds were in the form of iodized salts, but we observed that changing the iodized salt with a bromide salt did not change activity of the compound with 12 carbons (compare mr20350 and mr30868 in Table 1). This suggests that the nature of the counterion associated with the bis-imidazoline compounds does not influence their capacity to bind CXCR4.

We next investigated whether the bis-imidazoline compounds also bind to CXCR7/ACKR3, which we and others have identified as a second high-affinity receptor for CXCL12 [4,5] playing important roles in normal physiology and diseases [53,54]. We measured the ability of these compounds to bind to HEK293 cells stably expressing ST-ACKR3/CXCR7 in HTRF-based displacement binding experiments of red-CXCL12 used at 3 nM, a concentration close to its K_D_ value for the receptor [51]. As shown in Figure 1B, the bis-imidazoline compounds only marginally inhibited the HTRF signal, suggesting that they bind weakly to ACKR3 and as a consequence, do not inhibit CXCL12 binding to the receptor. Taken together, these data suggest that the bis-imidazoline compounds with an alkyl chain length of 8, 10 or 12 carbons at submicromolar concentrations can bind CXCR4 but not CXCR7/ACKR3. The three molecules, mr20347, mr20349 and mr20350, were further studied in the rest of the study.

### 2.3. Bis-Imidazoline Compounds Behave as CXCR4 Antagonists That Inhibit CXCL12 Signaling

To further characterize the capacity of bis-imidazolines to bind to CXCR4, we ran additional competition binding assays using the anti-CXCR4 monoclonal antibody (mAb) 12G5 as tracer and the A3.01 human T-cell line that endogenously expresses the receptor. This mAb binds CXCR4 with high affinity (K_D_ = 0.63 ± 0.03 nM, data not shown) and recognizes a conformational epitope on the receptor, which encompasses its N-terminus and extracellular loops 2 and 3 and overlaps the CXCL12 binding site [55]. A3.01 cells were incubated with 0.5 nM 12G5 in the presence or absence of increasing concentrations of mr20347, mr20349 or mr20350, or their counterparts with a linker with an odd number of carbons (mr20348 and mr34200). For comparison, experiments were also performed using AMD3100, CXCL12 or its antagonist analog P2G [56] as competitor. The amount of 12G5 that remained bound to A3.01 cells was then determined by flow cytometry using a phycoerythrin-conjugated goat anti-mouse Ig secondary antibody (Figure 2A and Table 1). The obtained competition curves led to the determination of IC_50_ values for CXCL12, AMD3100 and P2G of 6.3 ± 2.0, 84 ± 10 and 107 ± 10 nM, respectively. Posing that the bis-imidazoline molecules inhibit 12G5 binding by a competitive mechanism, and using the Cheng and Prusoff equation [57], we inferred affinity constants K_I_ for CXCR4 binding of 3.15, 42.0 and 53.5 nM, respectively, in the range of earlier data [51,56,58]. Confirming binding to CXCR4, the bis-imidazoline compounds also displaced 12G5 binding, though with an overall 20-fold reduced potency compared to AMD3100 and P2G. In this assay, the influence of alkyl chain length was less marked than in the CXCL12 binding competition assay (Figure 1A), although mr34200 with an 11-carbon chain consistently showed lower inhibitory potency. In contrast, bis-imidazoline compounds failed to block the binding of anti-CCR5 mAb 2D7 on A3.01 cells expressing the related chemokine receptor CCR5 (A3.01R5 cells), in contrast to the CCR5 chemokine CCL4 (Figure 2B), further supporting their specificity for binding to CXCR4.

We next investigated whether the bis-imidazoline compounds act as CXCR4 agonists or antagonists. To this end, we tested their capacity relative to CXCL12 to trigger chemotaxis of A3.01 cells using a transwell system previously described [24]. As shown in Figure 2C, CXCL12 induced a typical bell-shaped chemotactic response with maximal migration of A3.01 cells at 10 nM. In contrast, over a concentration range up to 10 μM, bis-imidazoline derivatives did not induce cell migration, indicating that they are not agonists for CXCR4. However, we validated that they effectively bind to CXCR4 by showing their inhibitory effect on chemotaxis of A3.01R5 cells induced by 10 nM CXCL12 (with inhibitory efficacies of 92%, 53% and 75% for mr20347, mr20349 and mr20350, respectively) (Figure 2D). This inhibition was unlikely due to altered cell integrity, because incubation of cells with 10 μM of bis-imidazoline compounds did not change the proportion of cells that are permeable to propidium iodide, compared to untreated cells (Figure 2E). The compounds also did not or only slightly (mr20350) decrease the migration of A3.01R5 cells induced by the CCR5 agonist PSC-RANTES/CCL5 (Figure 2F). These results therefore indicate that the bis-imidazoline compounds studied here are functional antagonists of CXCR4/CXCL12 signaling.

We and others previously showed that CXCR4/CXCL12-induced chemotaxis depends on activation of heterotrimeric G_i/o_ proteins and interaction with β-arrestins [25]. Our results here therefore suggest that bis-imidazolines may be ineffective for the mobilization of either of these proteins. However, it is now well established that GPCR ligands known as biased agonists can behave as antagonists for some signaling pathways (e.g., G-protein dependent) while being agonists for others (β-arrestins) [59,60]. A paradigmatic example of this is CXCL12 when it binds to CXCR7/ACKR3 [61]. It has also been reported that CXCR4 can activate other G-protein types and G-protein-independent pathways [1,19,62], and can have different properties depending on whether it exists as monomer, dimer or higher-size oligomer or as nanoclusters [63,64,65,66]. These data leave open the possibility that bis-imidazolines may have signaling properties or influence CXCL12 functions differently depending on the nature of the CXCR4 signaling pathway(s).

### 2.4. Bis-Imidazoline Compounds Target Acidic Residues of the CXCR4 Binding Pocket

To better characterize the CXCR4 binding pocket of the bis-imidazoline derivatives at the molecular level, we then combined docking experiments with site-directed mutagenesis. Docking for the compounds with an alkyl chain length of 8 (mr20347) or 12 carbons (mr20350) was studied on CXCR4 models built from the complexes of the receptor with the antagonist small molecule IT1t (PDB ID: 3ODU, [66]), the cyclic peptide antagonist CVX15 (PDB ID: 3OE0, [66]) or the viral chemokine antagonist vMIP-II (PDB ID: 4RWS, [67]). The 3D models of compounds were built and protonated on both imidazole’s rings in agreement with the ChemAxon software prediction (pH = 7.4). To dock the compounds, we used the SwissDock server and applied the blind docking approach during which ligand binding poses in the vicinity of all CXCR4 chemokine receptor cavities were generated.

In the published structures, the solved CXCR4 3D complexes showed a common ligand binding region close to the extracellular surface and more or less extended depending on ligand. Three sites were annotated in this ligand binding region: chemokine recognition site 1 in proximity of CXCR4 N-terminus outside of the receptor (CRS1), chemokine recognition site 2 in TM bundle (CRS2), bridged by a short intermediate region, CRS1.5. Further, CRS2, situated in the transmembrane region, is subdivided into two binding pockets: major subpocket and minor subpocket (Figure 3A). In the structure co-crystallized with vMIP-II, CRS1 that involves CXCR4 N-terminal residues binds the chemokine core, while CRS2, more particularly its minor subpocket, makes contact with the distal part of the chemokine N-tail. A similar situation was predicted in the recently published model of the CXCR4 complex with CXCL12 [68]. IT1t also mainly localizes into CRS2’s minor pocket, making hydrogen bonds with Asp97^2.63^ (superscript refers to the Ballesteros–Weinstein nomenclature) and Glu288^7.39^, while peptide antagonist CVX15 binds to CRS2’s major subpocket and occupies a more superficial position. In the different poses of bis-imidazoline compounds generated by SwissDock, the positively charged imidazole rings were placed near the negatively charged residues of both CRS2 subpockets (Appendix A), forming hydrogen bonds between imidazole nitrogen atoms and carboxylate oxygen atoms of CRS2 acidic residues. The flexible alkyl chain was then folded in such a way that the two imidazole rings were positioned in close proximity to CRS2 acidic residues, with each likely to interact with one of them. Although some differences in the poses generated are notable according to the CXCR4 structure used as a starting template (Appendix A), it was apparent that the imidazole rings form H-bonds with Asp97^2.63^ and Glu288^7.39^, similar to IT1t and the N-tails of chemokines. Along the docking prediction, other CXCR4 residues could also be predicted in the binding of bis-imidazolines (Appendix A), in particular, the acidic residues Glu32^1.26^, Asp171^4.60^ (similarly to CVX15) and Asp187^ECL2^ (similarly to CVX15 and vMIP-II), and Gln200^5.39^, a residue linking TM5 and TM6 and which is differently positioned in the complexes with CVX15 and IT1t [69].

We next explored in more detail the role of Asp97^2.63^, Asp171^4.60^, Asp262^6.58^ and Glu288^7.39^ in the binding of bis-imidazoline compounds through mutagenesis experiments. Although our models did not show a role for Asp262^6.58^ in the binding of bis-imidazolines, it is an interacting residue common to CVX15 and vMIP-II. Wild-type CXCR4 or the D97N, D171N, D262A or E288Q variant receptors were transiently expressed in the human astroglioma U373MG cells, which do not express endogenous CXCR4, and the capacity of mr20347 (n = 8), mr20349 (n = 10) or mr20350 (n = 12) to bind transfected cells was assessed through displacement experiments of 12G5 mAb binding (Figure 4). For comparison, experiments with CXCL12 or AMD3100 as competitor were also conducted, as binding of these molecules to the selected CXCR4 mutants has already been reported [70,71]. Saturation binding experiments of 12G5 confirmed that the CXCR4 constructs bind the mAb with similar affinities, compared to WT-CXCR4, with the K_D_ values ranging between 0.14 (D97N) and 1.4 (D171N and D262A) nM (Figure 4A). These results are in accordance with previous studies [70,71]. The maximum levels of binding however differed between the different CXCR4 constructs, which may be explained by the fact that these receptors do not have the same expression levels at the cell surface [70,72] and/or adopt different antigenic conformations [73].

We then measured the binding of a subsaturating concentration of 12G5 (1 nM) in the presence or absence of excess CXCL12 (1 μM), AMD3100 (10 μM) or bis-imidazoline compounds (10 μM). The results in Figure 4B show that bis-imidazolines, regardless of their alkyl chain length, lose their inhibitory capacity of 12G5 binding on D97N and E288Q mutants, confirming the molecular docking data indicating direct interactions between the imidazole moieties and CXCR4 CRS2’s Asp97^2.63^ and Glu288^7.39^. The binding of AMD3100 was similarly impaired by both mutations, in agreement with earlier works [71,74], while CXCL12 binding was sensitive to the mutation at Asp97^2.63^ [70] but substantially less to that at Glu288^7.39^ [74]. We confirmed that the mutations at Asp171^4.60^ and Asp262^6.58^ faintly changed CXCL12 binding [58,70] but decreased that of AMD3100 [71,75]. Interestingly, the D171N substitution inhibited mr20349 but not mr20347 or mr20350 in their capacity to compete with 12G5. This suggests that the Asp171^4.60^ contribution to binding of bis-imidazolines might depend upon their alkyl chain length. Finally, D262A mutation did not change the binding of these compounds, in agreement with the molecular docking experiments.

Considered altogether, these data indicate that the binding of bis-imidazolines to CXCR4 is primarily mediated by electrostatic interactions between the nitrogen atoms of imidazole rings and Asp97^2.63^ and Glu288^7.39^ within CRS2. Therefore, the docking poses of bis-imidazolines interacting at the same time with these two acidic residues were selected as their interaction mode in the CXCR4 receptor (see Figure 3B). Additional interactions may also be engaged with other acidic residues of the binding cavity (e.g., Asp171^4.60^) depending on the alkyl chain length. Interestingly, the docking experiments further indicated that to bind optimally to the carboxyl group of the acidic residues, i.e., to engage two hydrogen bonds with both oxygen atoms at the same time, the aromatic cycles of the derivatives must have two hydrogen bond donors on the same side at a distance of approximately 2.2Å, which is the case for the imidazole cycle. This explains the lack of affinity observed for the derivatives with pyrimidine instead of imidazole cycles (see above).

### 2.5. Assessment of Bis-Imidazoline Derivatives as Anti-HIV-1 Molecules

Because some acidic residues in the CXCR4 binding pocket including Asp97^2.63^, Asp171^4.60^ and Glu288^7.39^ are required for the receptor to act as an HIV-1 coreceptor, we sought to investigate whether bis-imidazolines could inhibit HIV-1 entry into target cells. To this end, we inoculated primary CD4 + T lymphocytes from healthy donors with virus clones pseudotyped with CXCR4-using (i.e., X4-tropic) envelope glycoproteins (Envs) in the presence or absence of varying concentrations of CXCL12, P2G, AMD3100 or bis-imidazoline compounds mr20347 (n = 8), mr20349 (n = 10) or mr20350 (n = 12). Infectivity levels were measured 48 h later by quantifying the activity of *Renilla* luciferase used as a reporter protein in infected cells. Two Env types were used, NL4-3 and X4-28, which we recently showed to be sensitive and resistant to inhibition by CXCL12, respectively [43]. As illustrated in Figure 5**,** CXCL12 and its antagonist variant P2G more potently inhibited infection with NL4-3 Env (Figure 5A) than with X4-28 (Figure 5B). A similar trend was observed with AMD3100, albeit less pronounced, which is consistent with our previous work [43] and other observations that HIV-1 strains may be cross-resistant to CXCL12 and AMD3100 [76], although AMD3100 retained by far the most potent antiviral activity of all tested molecules. As previously observed, P2G showed weaker anti-HIV-1 potency than CXCL12 due to its inability to internalize CXCR4 [51,56]. Regarding bis-imidazolines, anti-HIV-1 activity was observed with mr20349 and mr20350 but not with mr20347, suggesting a role for alkyl chain length. Interestingly, unlike the other inhibitory molecules, mr20349 and mr20350 more potently inhibited X4-28 than NL4-3 (extrapolated IC_50_ = 1.1 µM vs. 30 µM and 0.72 µM vs. 2.6 µM, respectively). These molecules at 1μM concentration actually inhibited X4-28 as efficiently as CXCL12 (Figure 5C). These molecules, however, did not or only slightly influence infection of CD4+ T lymphocytes by viruses pseudotyped with a R5-tropic Env (i.e., using CCR5 but not CXCR4 as coreceptor) (Figure 5D) or with the vesicular stomatitis envelope glycoprotein VSV-G that allows virus entry in a coreceptor-independent manner (data not shown).

Overall, these data indicate that binding of bis-imidazolines to CXCR4 may inhibit HIV-1 entry into target cells. However, this appears to be strain-dependent, as were the effects of CXCR4 mutations at Asp97^2.63^, Asp171^4.60^ and Glu288^7.39^ [70]. Our results also highlight the crucial role of the length of the alkyl linker in the regulation of anti-HIV-1 potency.

## 3. Materials and Methods

### 3.1. Ethics Statement

Primary CD4+ T lymphocytes were isolated from blood samples of anonymous healthy donors purchased from Etablissement Français du Sang (EFS), the French National Blood Agency. The donors provided written informed consent to EFS at the time of blood collection. The pseudotyped viruses used in the present study were previously described [43,77].

### 3.2. Cells, Reagents and Plasmids

A3.01, A3.01R5 and U373MG cells were obtained and cultured as described [70,78]. HEK293 cells stably expressing Lumi4-Tb-labeled SNAP-tagged CXCR4 or CXCR7 were purchased from Cisbio Bioassays (Codolet, France) (hereafter referred to as Tag-lite CXCR4 or CXCR7 cell lines). Cells were immediately used after quick thawing at 37 °C and suspension in the Tag-lite labeling medium (Cisbio Bioassays) as described [50]. Human CD4+ T lymphocytes were purified from PBMCs of healthy blood donors by centrifugation on Ficoll-hypaque density gradient (PAA laboratories, Pasching, Austria) followed by immunomagnetic positive selection using CD4 MicroBeads (Miltenyi Biotec, Bergisch Gladbach, Germany). They were then maintained for 2 days in phytohemagglutinin (PHA) (1 µg/mL) and interleukin-2 (IL-2) (300 IU/mL)-containing RPMI-1640 complete medium, and then for an additional 3 days in the presence of IL-2 alone, as previously described [43,77]. MVC and AMD3100 were obtained from Sigma-Aldrich (St. Louis, USA) and CXCL12 from Miltenyi Biotec. P2G-CXCL12 and PSC-CCL5 were provided by Dr F. Baleux (Institut Pasteur, Paris, France) and Dr O. Hartley (University of Geneva, Geneva, Switzerland), respectively. The plasmids and methods used for producing the *Renilla* luciferase reporter virus clones were previously described [43,77]. The plasmids applied for the mapping experiments using the 12G5 mAb and their transfection in U373MG cells were also previously described [70].

### 3.3. Screening of the CERMN Chemical Library

All compounds are in DMSO at a concentration of 3.3 mg/mL (average concentration of 9.85 ± 3.7 mM). They were screened at a 1:500 dilution and a final DMSO concentration of 2.5% (*v*/*v*). The assay was performed into white, small volume, HiBase, medium-binding bar-coded 384-well plates (Greiner Bio-One, Gallen, Switzerland). For each plate, columns 1, 2, 23 and 24 were dedicated to negative (16 wells in the presence of AMD3100 at a final concentration of 15 µM) and positive (16 wells in the presence of DMSO alone) controls, where the same amount of DMSO was applied. Initially, 1 μL of compound in DMSO solution was loaded into dry wells from columns 3 to 22. Then, 9 μL of red CXCL12 (20 nM final concentration) and 10 µL of Tag-lite CXCR4 stable cell line (5000 cells per well) were added sequentially to all wells. After a 2 h incubation at 16 °C, the fluorescence emission was measured at 620 nm and 665 nm using an excitation wavelength at 343 nm on a Safire2 (Tecan, Männedorf, Switzerland) microplate reader. The HTRF ratio was defined as following: Ratio = (Fluorescence intensity at 665 nm/Fluorescence intensity at 620 nm) × 10^4^. The relative HTRF ratio of each compound was calculated via the following equation: relative HTRF ratio = HTRF ratio of the compound/HTRF ratio value of the negative controls. For each plate, the Z′-factor [52] was calculated as follows: Z′ = 1 − (3σ_c+_ + 3σ_c−_)/(|μ_c+_ − μ_c−_|), where σ_c+_ and σ_c−_ are the data standard deviation for the high reference control and low reference control, respectively, and |μ_c+_ − μ_c−_| is the absolute value of the difference of the two control signal means.

### 3.4. HTRF Dilution Series Assay on CXCR4 and CXCR7

A three-fold serial dilution in DMSO was performed for each compound. Each concentration was tested in duplicate. For these experiments, Tag-lite CXCR4 and Tag-lite CXCR7 cell lines (5000 cells per well) were used. For both cell lines, the assay and the measurements were performed as described for the CERMN chemical library screening (Section 3.1): the only difference was the final CXCL12 concentration for CXCR7 of 3 nM (instead of 20 nM for CXCR4). The IC_50_ values were determined with the Prism Software version 9 (Graph-Pad Software, San Diego, USA) using a one-site competitive binding model.

### 3.5. Antibody Binding Experiments

Saturation binding experiments of anti-CXCR4 mAb 12G5 were performed as follows. Cells (5 × 10^4^ A3.01 cells or 2.10^5^ U373MG cells) in conical 96-well plates were incubated at room temperature for 30 min with varying concentrations of unconjugated 12G5 (BD Biosciences, San Jose, USA) in PBS containing 2% bovine serum albumin and 0.1% NaN_3_. Non-specific binding (NSB) was determined in parallel in the presence of 10 μM AMD3100 or using control IgG as ligand. Cells were then washed once with ice-cold PBS and further incubated at 4 °C for 30 min with phycoerythrin-conjugated goat anti-mouse Ig secondary antibody (BD Biosciences). Cells were then washed twice with ice-cold PBS and fixed in PBS containing 2% paraformaldehyde. Mean Fluorescence Intensities (MFI) as a readout of 12G5 binding to cells were measured by flow cytometry using a FACSCantoII^TM^ (BD Biosciences). Specific 12G5 binding to CXCR4 was then calculated by subtracting NSB from the total binding of the mAb. Competition binding experiments in the presence of CXCR4 inhibitors were performed in the same manner using unconjugated 12G5 at 1 nM. The IC_50_ values for half-maximal inhibition of mAb binding by the inhibitors were determined with the Prism Software using a one-site competitive binding model. The dissociation constants K_I_ for the inhibitors were calculated according to the Cheng and Prusoff equation [57]: K_I_ = [IC_50_/(1 + L/K_D_)], where L and K_D_ represent the mAb concentration and the dissociation constant of the mAb-receptor complex, respectively. Binding of unconjugated 2D7 (BD Biosciences) to A3.01R5 cells was determined, as described above using Alexa Fluor 647-conjugated goat anti-mouse IgG (ThermoFisher, Waltham, USA) at a 1:500 dilution in binding buffer.

### 3.6. Chemotaxis Assays

Chemotaxis of A3.01 or A3.01R5 cells was determined using a Transwell system as previously described [79]. Cells (1.5 × 10^5^) in prewarmed RPMI-1640 medium supplemented with 20 mM HEPES and 1% human serum AB were added to the top chambers of HTS-Transwell-96 Well Permeable Supports with polycarbonate membrane of 5 μm pore size (Corning, New York, USA). The same medium with or without (spontaneous migration) CXCL12, PSC-CCL5 or bis-imidazoline compounds was added to the lower chambers. Inhibition assays of chemokine-mediated chemotaxis by AMD3100, MVC, P2G or bis-imidazolines were performed by adding inhibitors to both the upper and lower chambers. Cells were allowed to migrate for 5 h at 37 °C in humidified air with 5% CO2. The number of cells migrating across the polycarbonate membrane was then assessed by flow cytometry with a FACS CantoII (BD Biosciences).

### 3.7. Cell Cytotoxicity Assay

All compounds were tested for cytotoxicity at a 10 µM concentration in A3.01 cells (1 × 10^5^). Cells were incubated ON at 37 °C in complete RPMI medium and then washed twice in FACS buffer. Cell viability was assessed by the propidium iodide exclusion method, staining dead cells, by flow cytometry with a FACS CantoII (BD Biosciences).

### 3.8. Infection Inhibition Assays

PHA/IL-2 activated CD4+ T cells in round-bottom 96-well plates (2 × 10^5^ cells per well) were infected with similar quantities of the different viruses (10–20 ng of the viral protein Gag p24) in the presence or absence of increasing concentrations of inhibitors. Infection was undertaken in the presence of IL-2 at 300 IU/mL. At 48 h post-infection, cells were washed once in PBS, lysed and infectivity was determined by measuring *Renilla* luciferase activity (Renilla Luciferase Assay System, Promega, Madison, USA) using a 96-well plate lumi/fluorimeter Mithras LB940 (Berthold, Bad Wildbad, Germany).

### 3.9. Docking Analysis

The initial model of mr20347 (alkyl chain length = 8) and mr20350 (alkyl chain length = 12) was built using the Biovia Discovery Studio and the compounds’ protonation state at pH 7.4 was predicted using standard tools of the ChemAxon Package (http://www.chemaxon.com). The majority microspecies protonated on imidazole nitrogen at this pH was used for docking studies of each compound.

The crystallographic coordinates of human CXCR4 used in this study were obtained from three X-ray structures available in PDB databank: (i) from the antagonist IT1t/CXCR4 complex (PDB ID: 3ODU, a structure refined to 2.5 Å with a free R factor of 28.2% [66]); (ii) from the CVX15 cyclic peptide antagonist/CXCR4 complex (PDB ID: 3OE0, a structure refined to 2.9 Å with a free R factor of 26.7% [66]); and (iii) from the vMIP-II viral antagonist/CXCR4 complex (PDB ID: 4RWS, a structure refined to 3.1 Å with a free R factor of 27.4% [67]). As the CXCR4 constructs used for the crystallization of CXCR4/It1t and CXCR4/CVX15 complexes contained a T4 lysozyme (T4L) fusion inserted between TM 5 and TM6, the T4 lysozyme fusion protein was removed and the third intracellular loop was rebuilt (@tome-2 server). Further, to the thermostabilizing L125^3.41^W mutation present in all three X-ray structures, two other residues were mutated in the chemokine receptor structure of 4RWS: T240^6.36^P and D187^ECL2^C. Therefore, the mutated residues were rebuilt in agreement with the CXCR4 sequence (Uniprot entry: P61073). The hydrogen atoms were added on all CXCR4 3D models.

Docking was performed using the EA Dock DSS docking software (http://www.swissdock.ch/) [80] through the SwissDock server [81]. In a nutshell, the binding modes are generated on all found binding cavities, with the CHARMM22 force-field used to compute the binding energies. The best interacting modes are then post-processed using the implicit solvation model FACTS (Fast Analytical Continuum Treatment of Solvation) [82] to include the desolvation effect of ligand binding. Finally, the clustering of the binding modes is performed using a 2 Å radius.

### 3.10. Compounds’ Synthesis

All commercially available compounds were used without further purification. NMR spectra were recorded at 400 MHz (Bruker Avance III 400 MHz, Billerica, USA) for ^1^H NMR, and at 100 MHz for ^13^C NMR in DMSO-*d6* with chemical shifts (δ) given in ppm relative to TMS as internal standard. Standard abbreviations are used to describe peak splitting patterns. Coupling constants *J* are reported in Hertz (Hz). High-resolution mass spectra (HRMS) were obtained by electrospray (ESI, sampling cone 50 V, capillary 0.3 kV) on a Xevo G2-XS Qtof Waters mass spectrometer (Waters Corp, Milford, USA).

Compounds mr20345, mr20346, mr20347, mr20348, mr20349, mr20350 and mr30868 were prepared according to the known literature procedure [83,84] (see Figure 1).

The analog procedure was applied to synthetize mr34200 (*N*-(2,5-dihydro-*1H*-imidazol-2-yl)-*N′*-(4,5-dihydro-1*H*-imidazol-2-yl) undecane-1,11-diamine dihydrobromide) (Schema 1). Then, 0.50 g (2.68 mmol, 1.0 equiv) of undecane-1,11-diamine and 1.32 g (5.36 mmol, 2 equiv.) of 1-(4,5-dihydro-1*H*-imidazol-2-yl)-3,5-dimethyl-pyrazole hydrobromide in acetonitrile (30 mL) were refluxed for 6 h. The cold solution was evaporated under reduced pressure and the residue was dissolved in a minimum amount of methanol, and filtered. Diethyl ether was added to this solution and the mixture was kept overnight at 0 °C. The precipitate was collected, extensively washed with diethyl ether and dried under vacuum to afford the product as beige gum (0.55 g, 43%).

^1^H NMR (400MHz, DMSO-*d_6_*) δ = 8.15 (t, *J* = 5.9 Hz, 2H), 7.95 (bs, 2H), 3.59 (bs, 8H), 3.15 (q, *J* = 6.7 Hz, 4H), 1.48 (q, *J* = 6.8 Hz, 4H), 1.30–1.25 (m, 14H).

^13^C NMR (100 MHz, DMSO-*d_6_*) δ = 159.9 (2C), 42.9 (4C), 42.8 (2C), 29.4 (2C), 29.3, 29.2 (2C), 29.0 (2C), 26.5 (2C).

HRMS (m/z) calcd for C_17_H_35_N_6_ [M+H] + 323.2923, found: 323.2923.

## 4. Conclusions

Through an HTRF-based high-throughput screening of the CERMN chemical library and functional assays, we identified bis-imidazoline compounds as new CXCR4 antagonists. These molecules seemed to be specific to CXCR4 and, in particular, did not bind, or only marginally bound, the related receptors CXCR7/ACKR3 and CCR5. These compounds are formed by two imidazole moieties, which are linked by an alkyl chain and lodge in CXCR4’s CRS2, in particular, its minor subpocket. These compounds showed inhibitory capacities of CXCL12 binding and signaling and of HIV-1 infection that varied with the alkyl chain length. The compounds with a chain length of 8, 10 or 12 carbons were more potent than those with a shorter chain or with an odd number of carbons. Interestingly, the compounds with a chain length of 8, 10 or 12 carbons also differentially influenced different functions of the receptor. For instance, mr20347 (n = 8) more potently inhibited CXCL12 binding and CXCL12-mediated chemotaxis than mr20350 (n = 12) but had no effect on HIV-1 infection. In contrast, mr20350 showed the strongest anti-HIV-1 activity. Mr20347 was also globally less potent in displacing the binding of mAb 12G5. Molecular modeling and docking combined with site-directed mutagenesis suggested that the alkyl chain adopts a conformation allowing the localization of both imidazole groups in CRS2’s minor subpocket. Varying the length of the alkyl chain length subtly changes the position of imidazole groups in CRS2, and the receptor’s acidic residues with which they interact, which may explain the differential effects on the different functions of CXCR4. Due to the openness and negative charge of its binding pocket, CXCR4 can bind a great variety of structurally different ligands. However, evolution testified CXCL12 as the ligand mediating most of the essential functions of CXCR4, raising concerns about the use of CXCR4 as a therapeutic target in diseases. The critical role of the CXCL12/CXCR4 couple is also attested by its high conservation within the animal kingdom, both structurally and functionally, especially between human and mouse (89 and 99% homology for CXCR4 and CXCL12, respectively), making mouse models very relevant for understanding the pathophysiology of human diseases as well as efficacy and safety of CXCR4-targeting therapies. Our results here highlight a yet unexplored way to modulate the positioning of functional groups in CXCR4’s CRS2 and in doing so, differentially alter distinct functions of the receptor, ideally diminishing some involved in diseases and preserving others physiologically important. Future studies in mouse models of CXCR4-associated diseases will be instrumental to validate this therapeutic potential of bis-imidazolines.

## Data Availability

All available data are presented in this article.

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
