# Peer review of "Discovery of Bis-Imidazoline Derivatives as New CXCR4 Ligands"

_molecules, 2023, doi:10.3390/molecules28031156_

Round 1
Reviewer 1 Report
In the present study, researchers targeted the CXCR4 receptor for its antagonists to modulates its specific functions and related pathways for treatment of different diseases. But, there are some questions for the manuscript to respond before publication.
1- There is no Structure activity relationship (SAR) analysis for candidate molecules to discuss active molecular groups to interact CXCR4 receptor.
2- All of candidate compounds count as antagonist for the receptor. Is there a possibility for the compounds act as agonist and enhance the receptor cascade? this point should be discussed.
3- It understood from the binding positions that the candidate compounds act as a competitive inhibitor of CXCL12 ligand. Is there any possibility for the candidate compound act as a non-competitive inhibitor via the modify receptor structure?
4- Authors specifically pointed out that the candidate drugs might be effective against HIV infection. Is there any similar drug approach to compare and confirm the mode of action in wet experiment?
5- Animal experimental models should be discussed for pros and cons of in vitro models for further confirmations.
Reviewer 2 Report
In this interesting work entitled “Discovery of bis-imidazoline derivatives as new CXCR4 ligands”, authors reported the discovery of a set of small molecules based on the bis-imidazoline scaffold as novel CXCR4 ligands with anti-HIV activity. The work is well-organized, and the findings are presented and addressed clearly. Furthermore, the experimental portion is sound and provides sufficient information and data. I think that the readers of the journal ought to find this work interesting. I recommend it for publishing after addressing the following points.
It would be nice if the author provide a context and information about imidazoline derivatives as antiviral agents in the introduction.
I wonder if authors could improve the quality of the figures.
In Results and discussion section; authors have to cite and mention the acquired results numerically in the text, rather than relying just on figures to present the results.
Regarding the synthesis of Compounds mr20345, mr20346, mr20347, mr20348, mr20349, mr20350 and mr30868, as well as the synthesis of mr34200 it will be better if the authors provide the synthetic schemes.
Please revise the following information:
HRMS (m/z) calcd for C17H36Br2N6 [M+H]+ 323.2923, found: 323.2923.
Reviewer 3 Report
The manuscript by Philippe Colin et al. identified the bis-imidazoline compounds as new CXCR4 antagonists. These molecules seemed to be specific to CXCR4 and in particular did not bind or only marginally bind the related receptors CXCR7/ACKR3 and CCR5.
These results are acceptable to publish in Molecules.
Author Response
We thank the reviewer for her/his positive appreciation of our work and for accepting the manuscript.
Round 2
Reviewer 1 Report
The authors answered the all of the concerns in the revision. I recommend it to be published in the present form.
Reviewer 2 Report
The manuscript has been improved, and should be accepted now.